

# Accumulation of sequence variants in genes of Wnt signaling and focal adhesion pathways in human corneas further explains their involvement in keratoconus

Justyna A. Karolak[1,2], Tomasz Gambin[3], Malgorzata Rydzanicz[4], Piotr Polakowski[5], Rafal Ploski[4], Jacek P. Szaflik[5] and Marzena Gajecka[1,2]

[1] Chair and Department of Genetics and Pharmaceutical Microbiology, Poznan University of Medical Sciences, Poznan, Poland
[2] Institute of Human Genetics, Polish Academy of Sciences, Poznan, Poland
[3] Institute of Computer Science, Warsaw University of Technology, Warsaw, Poland
[4] Department of Medical Genetics, Medical University of Warsaw, Warsaw, Poland
[5] Department of Ophthalmology, Medical University of Warsaw, Warsaw, Poland

Corresponding author
Marzena Gajecka,
gamar@man.poznan.pl

## ABSTRACT

**Background:** Keratoconus (KTCN) is a protrusion and thinning of the cornea, resulting in loss of visual acuity. The etiology of KTCN remains unclear. The purpose of this study was to assess the potential involvement of new genetic variants in KTCN etiology based on both the genomic and transcriptomic findings recognized in the same corneal tissues.

**Methods:** Corneal tissues derived from five unrelated Polish individuals with KTCN were examined using exome sequencing (ES), followed by enrichment analyses. For comparison purposes, the datasets comprising ES data of five randomly selected Polish individuals without ocular abnormalities and five Polish patients with high myopia were used. Expression levels of selected genes from the overrepresented pathways were obtained from the previous RNA-Seq study.

**Results:** Exome capture discovered 117 potentially relevant variants that were further narrowed by gene overrepresentation analyses. In each of five patients, the assessment of functional interactions revealed rare (MAF ≤ 0.01) DNA variants in at least one gene from Wnt signaling (*VANGL1*, *WNT1*, *PPP3CC*, *LRP6*, *FZD2*) and focal adhesion (*BIRC2*, *PAK6*, *COL4A4*, *PPP1R12A*, *PTK6*) pathways. No genes involved in pathways enriched in KTCN corneas were overrepresented in our control sample sets.

**Conclusions:** The results of this first pilot ES profiling of human KTCN corneas emphasized that accumulation of sequence variants in several genes from Wnt signaling and/or focal adhesion pathways might cause the phenotypic effect and further points to a complex etiology of KTCN.

## INTRODUCTION

Keratoconus (KTCN) is an eye disease characterized by progressive thinning and conical protrusion of the cornea (*Karolak & Gajecka, 2017*). The structural abnormalities in different layers of corneal tissue result in altered refractive powers and a loss of visual function (*Rabinowitz, 1998*). The first symptoms of KTCN usually appear during puberty or early in the third decade of life (*Rabinowitz, 1998*). The management of this condition depends on the disease stage and includes visual correction by contact lenses, corneal collagen cross-linking, or corneal transplant surgery (*Sarezky et al., 2017*; *Vazirani & Basu, 2013*). The incidence of KTCN is one in 2,000 individuals in the general population (*Rabinowitz, 1998*). However, the incidence may vary depending on the geographic location and ethnicity of the studied population. The estimated incidence of KTCN is higher in Indians, Chinese, Pacific, and Maori ethnicities compared with Caucasians (*Pearson et al., 2000*; *Georgiou et al., 2004*; *Kok, Tan & Loon, 2012*; *Patel & McGhee, 2013*; *Gokhale, 2013*; *Godefrooij et al., 2016*). The environmental factors, such as eye rubbing or contact lens wear, influence disease development (*Hashemi et al., 2019*). However, genetic triggers also play an important role in KTCN (*Karolak & Gajecka, 2017*; *Abu-Amero, Al-Muammar & Kondkar, 2014*). Several genomic strategies have been implemented for finding candidate genes, including both simple molecular techniques and high-throughput technologies (*Karolak & Gajecka, 2017*; *Abu-Amero, Al-Muammar & Kondkar, 2014*; *Bykhovskaya, Margines & Rabinowitz, 2016*; *Mas Tur et al., 2017*; *Valgaeren, Koppen & Van Camp, 2018*; *Loukovitis et al., 2018*).

Linkage studies, performed in KTCN families, have led to the identification of multiple chromosomal regions linked to KTCN, including two replicated loci at 5q (*Tang et al., 2005*; *Li et al., 2006*; *Bisceglia et al., 2009*; *Rosenfeld et al., 2011*; *Bykhovskaya et al., 2016*). The genome-wide association studies have detected variants mapped near *HGF*, *COL5A1*, *FOXO1*, *RAB3GAP1*, or *ZNF469*, associated with KTCN risk (*Li et al., 2012*; *Burdon et al., 2011*; *Hoehn et al., 2012*; *Lu et al., 2013*). However, their contribution to KTCN needs to be clarified. Several candidate genes for KTCN have also been identified using Sanger sequencing or next-generation sequencing including *VSX1*, *SOD1* and *DOCK9* or *SKP1*, *MPDZ*, *FLG*, *PPIP5K2*, and *PCSK1*, respectively (*Héon et al., 2002*; *Karolak et al., 2015*, *2017*; *Udar et al., 2006*; *Lucas et al., 2018*; *Khaled et al., 2019*; *Magalhães et al., 2019*). However, variants detected in those genes were present in a small fraction of KTCN patients or particular populations only.

Analysis of transcriptome profiles of human KTCN and non-KTCN corneas by a high-throughput RNA sequencing (RNA-Seq) showed the deregulation of numerous genes in KTCN corneas (*Kabza et al., 2017*). The significant downregulation was observed among genes in collagen synthesis and maturation pathways, as well as in the TGF-β, Hippo and Wnt signaling pathways (*Kabza et al., 2017*). The results of subsequent RNA studies further support the potential role of genes involved in the extracellular matrix, TGF-β and Wnt molecular cascades in KTCN pathogenesis (*Khaled et al., 2018*;

*You et al., 2018*; *Sharif et al., 2019*). Since these signaling pathways influence the corneal organization and play a role in the regulation of extracellular matrix components, the genes encoding the core elements of the mentioned pathways were proposed as novel candidate genes for KTCN.

To assess the involvement of new genetic variants in KTCN etiology, we performed a further molecular investigation of corneas of Polish patients with KTCN, previously tested by RNA-Seq, using exome sequencing (ES) approach.

## MATERIALS AND METHODS

### Patients

All patients with KTCN underwent a complete ophthalmic evaluation in the Department of Ophthalmology, Medical University of Warsaw, Poland. The KTCN diagnosis was made based on the criteria previously described (*Kabza et al., 2017*; *Karolak et al., 2016*). The study protocol was approved by the Institutional Review Board at Poznan University of Medical Sciences (453/14; 755/19). All individuals provided informed consent after the possible consequences of the study were explained, in accordance with the Declaration of Helsinki.

### Material collection and DNA extraction

The pairs of whole corneal tissues and blood samples were obtained from previously evaluated five KTCN patients (KC15, KC16, KC17, KC18, and KC19) (*Kabza et al., 2017*) undergoing a penetrating keratoplasty procedure. The clinical characteristics of these patients are presented in Table S1. Genomic DNA samples from the corneas were extracted using the Cells and Tissue DNA Isolation Kit (Norgen Biotek, Thorold, ON, Canada) according to the manufacturer's protocol. Genomic DNA samples were isolated from the blood lymphocytes using the Gentra Puregene Blood Kit (Qiagen, Hilden, Germany), as previously described (*Karolak et al., 2016*).

### Exome sequencing

This pilot ES study was conducted with 50 ng of genomic DNA of whole corneal tissues (KC15, KC16, KC17, KC18, and KC19) using the SureSelectQXT Reagent Kit combined with the SureSelectXT Human All Exon V5 (Agilent Technologies, Cedar Creek, TX, USA) according to manufacturer's instruction. Prepared libraries were paired-end sequenced ($2 \times 100$ bp) on an Illumina HiSeq1500 (Illumina, San Diego, CA, USA). For each cornea sample > 80 million read pairs were generated resulted in $>100\times$ of mean coverage. Sequence readouts were initially analyzed with bcl2fastq software to generate reads in fastq format. These reads were mapped against a human genome reference sequence (GRCh37) using the Burrows–Wheeler Alignment (BWA), which was followed by BAM post-processing and variant calling using HaplotypeCaller and the GATK suite (*McKenna et al., 2010*). Finally, ANNOVAR (*Wang, Li & Hakonarson, 2010*) was used to annotate relevant information about gene names, predicted variant pathogenicity, reference allele frequencies and metadata from external resources and then to add these to the variant call format (VCF) file.

### Variants selection

Sequence variants identified in KTCN corneas were filtered in a step-wise manner to exclude synonymous variants and variants with minor allele frequency (MAF) greater than 0.01 in our internal exome database (DMG) consisting of 3,000 Polish individuals, ExAC Browser (http://exac.broadinstitute.org/), GnomAD database (https://gnomad.broadinstitute.org/), and the 1,000 Genomes Project (http://www.1000genomes.org). Moreover, variants predicted as neutral by MutationTaster (*Schwarz et al., 2010*), PolyPhen-2 (*Adzhubei, Jordan & Sunyaev, 2013*), LRT (*Chun & Fay, 2009*) and SIFT (*Kumar, Henikoff & Ng, 2009*) tools were filtered out. The additional exclusion criterion was negative conservation scores in PhyloP (*Pollard et al., 2010*) analysis.

### Sanger sequencing

To confirm variants detected by ES in five KTCN corneas, Sanger sequencing of DNA in matching blood samples was performed. Briefly, fragments of genes containing particular variants were amplified using *Taq* DNA Polymerase (Thermo Scientific, San Jose, CA, USA). Following purifications with the use of FastAP Thermosensitive Alkaline Phosphatase and Exonuclease I (Thermo Scientific, San Jose, CA, USA), the amplicons were sequenced using BigDye Terminator v3.1 Cycle Sequencing Kit (Applied Biosystems Inc., Foster City, CA, USA). Samples were analyzed on the ABI Prism 3730xl genetic analyzer (Applied Biosystems Inc., Foster City, CA, USA). Sequences were assembled using Sequencher 5.0. Software (GeneCodes Corporation, Ann Arbor, MI, USA).

### Pathway overrepresentation analysis

The overrepresentation analysis of molecular pathways among genes with identified sequence variants, which met the filtering criteria, was performed using the ConsensusPathDB tool (*Kamburov et al., 2013*). Only pathways with $p$-value $\leq 0.01$ and sharing at least two genes with our gene set were analyzed.

### Ethnically-matched control datasets

For comparison purposes, the datasets comprising ES data of five randomly selected Polish individuals without ocular abnormalities and five Polish patients with high myopia (HM) were used. Variants from control ES data were selected using the same filtering criteria as we used for the KTCN study, followed by enrichment analyses.

### Expression analysis

Expression levels of selected genes from overrepresented pathways, given in the gene-level transcripts per million (TPM), were obtained from the RNA-Seq study, which has been previously performed in the same corneal material (*Kabza et al., 2017*). The expression values of the particular genes were compared between corneal samples in which variant(s) was identified and in samples without the analyzed variant.

## RESULTS

This pilot ES screening of KTCN corneas revealed 117 potentially relevant variants with MAF $\leq 0.01$ in our internal DMG cohort and public databases and fulfilling all of the remaining filtering criteria (Table S2). Genes with identified rare nucleotide variants were

**Table 1 Top pathways overrepresented across the genes with identified sequence variants in KTCN corneas detected by ConsensusPathDB server (*p*-value cutoff 0.01).**

| Pathway name | Source | Genes[a] | *p*-Value |
|---|---|---|---|
| Disassembly of the destruction complex and recruitment of AXIN to the membrane | Reactome | *FZD2, LRP6, WNT1* | 0.000873181 |
| Wnt signaling | Wikipathways | *VANGL1, WNT1, PPP3CC, LRP6, FZD2* | 0.000893629 |
| Type I hemidesmosome assembly | Reactome | *PLEC, COL17A1* | 0.001477943 |
| Wnt signaling in kidney disease | Wikipathways | *WNT1, LRP6, FZD2* | 0.001788119 |
| Wnt signaling pathway - *Homo sapiens* (human) | KEGG | *VANGL1, WNT1, PPP3CC, LRP6, FZD2* | 0.002830193 |
| RAB GEFs exchange GTP for GDP on RABs | Reactome | *ANKRD27, DENND1A, RABGEF1, DENND4C* | 0.002959219 |
| The activation of arylsulfatases | Reactome | *ARSD, ARSB* | 0.003148218 |
| Epithelial to mesenchymal transition in colorectal cancer | Wikipathways | *WNT1, COL4A4, LRP6, FZD2, MEF2D* | 0.003952926 |
| cGMP-PKG signaling pathway - *Homo sapiens* (human) | KEGG | *ATP2A1, PIK3CG, PPP1R12A, PPP3CC, MEF2D* | 0.004277236 |
| MicroRNAs in cardiomyocyte hypertrophy | Wikipathways | *HDAC9, FZD2, LRP6, PIK3CG* | 0.004456905 |
| Wnt-beta-catenin signaling pathway in Leukemia | Wikipathways | *WNT1, LRP6* | 0.008974682 |
| Rab regulation of trafficking | Reactome | *ANKRD27, DENND1A, RABGEF1, DENND4C* | 0.009329868 |
| Focal adhesion | Wikipathways | *BIRC2, PAK6, COL4A4, PPP1R12A, PTK6* | 0.009582658 |
| Regulation of RAS by GAPs | Reactome | *NF1, SPRED2* | 0.009787951 |

**Note:**
[a] Analyzes were performed in genes with variants fulfilling the filtering criteria: (i) MAF ≤ 0.01 in our internal database, the ExAC Browser, GnomAD database and the 1,000 Genomes Project Variants; (ii) positive conservation scores in PhyloP; (iii) pathogenic/damaging in MutationTaster, PolyPhen2, LRT and SIFT tools.

enriched in 14 molecular pathways, including Wnt signaling (*VANGL1, WNT1, PPP3CC, LRP6, FZD2*) and focal adhesion (*BIRC2, PAK6, COL4A4, PPP1R12A, PTK6*) pathways (Table 1). The analysis of functional interactions between genes from overrepresented molecular pathways allowed for further narrowing the number of candidate KTCN variants (Table 2).

Sanger sequencing performed in blood samples derived from the five studied individuals confirmed all variants identified with the use of ES in matched corneas.

The overrepresentation analysis performed among genes filtered in a step-wise manner in control datasets revealed an enrichment in 16 (i.e., O-linked glycosylation, C-type lectin receptors, termination of O-glycan biosynthesis, nucleotide-binding oligomerization domain) and nine (i.e., cargo recognition for clathrin-mediated endocytosis VLDL clearance, glyoxylate and dicarboxylate metabolism, statin pathway DNA damage recognition in GG-NER) pathways in HM patients and individuals without ocular disease, respectively. No genes involved in pathways enriched in KTCN corneas were overrepresented in our control sample sets containing randomly selected Polish individuals without ocular abnormalities ($n = 5$) and Polish patients with HM ($n = 5$).

The TPM values of genes from overrepresented pathways in corneas of patients carrying particular gene variants and in KTCN individuals without the analyzed gene variation were calculated based on our previous RNA-Seq study data (Kabza et al., 2017). The expression values of the genes varied between patients with particular variants and the

**Table 2 The list of nonsynonymous sequence variants in genes from Wnt signaling and focal adhesion pathways, identified in KTCN corneas in ES.**

| Identifier | Gene | Acession number | Position[a] | cDNA | Protein | DMG[b] | GnomAD[c] | Rs_id[d] |
|---|---|---|---|---|---|---|---|---|
| KC15 | WNT1 | NM_005430.3 | chr12:49375373G>T | c.1063G>T | p.(V355F) | 0 | NA[e] | rs387907358 |
| | PTK6 | NM_005975.3 | chr20:62161533G>T | c.1066C>A | p.(P356T) | 0 | NA | NA |
| KC16 | FZD2 | NM_001466.3 | chr17:42636294G>A | c.1238G>A | p.(R413Q) | 0 | 0.00002396 | rs758351214 |
| | VANGL1 | NM_001172411.1 | chr1:116194075C>T | c.41C>T | p.(S14L) | 0 | NA | NA |
| | COL4A4 | NM_000092.4 | chr2:227924157C>T | c.2347G>A | p.(G783R) | 0.000333 | NA | rs1202230056 |
| KC17 | LRP6 | NM_002336.2 | chr12:12274080G>A | c.4822C>T | p.(P1608S) | 0 | NA | NA |
| | PPP3CC | NM_001243975.1 | chr8:22389795T>C | c.1199T>C | p.(M400T) | 0 | NA | NA |
| KC18 | BIRC2 | NM_001166.4 | chr11:102248388A>G | c.1528A>G | p.(I510V) | 0.001 | 0.00002148 | rs749829698 |
| KC19 | PPP1R12A | NM_001244992.1 | chr12:80266702T>C | c.254A>G | p.(N85S) | 0.000667 | 0.00002422 | rs370959842 |
| | PAK6 | NM_001276717.1 | chr15:40566454C>A | c.1855C>A | p.(P619T) | 0 | NA | NA |

**Notes:**
[a] Human Genome Browser—hg19 assembly (GRCh37).
[b] Minor allele frequency based on internal control exome database covering of 3,000 Polish individuals; Department of Medical Genetics (DMG), Medical University of Warsaw, Warsaw, Poland.
[c] Minor allele frequency based on GnomAD database v2.1.2.
[d] NCBI dbSNP Build 151.
[e] Not available.

mean expression values of the same genes in the KTCN individuals without these variants. The expression profile of genes from overrepresented pathways in KTCN corneas is presented in Table 3.

The data were deposited in the freely accessible ClinVar database (submission ID: SUB3758236 and SUB4926771).

# DISCUSSION

The rapid development of next-generation DNA sequencing methods has greatly improved the ability to detect genetic variations (Koboldt et al., 2013). However, KTCN, like many other ophthalmic diseases, displays genetic heterogeneity hindering the identification of a factor unambiguously influencing its development (Karolak & Gajecka, 2017; Stone et al., 2004).

The results of this pilot ES study of corneas obtained from five KTCN patients undergoing penetrating keratoplasty revealed various rare possibly pathogenic variants in genes that were overrepresented in several molecular pathways, including the Wnt signaling and focal adhesion. Interestingly, these pathways have been proposed as involved in KTCN etiology based on our previous RNA-Seq study, performed in experimental material derived from the same KTCN patients, as well as other experiments (Kabza et al., 2017; Khaled et al., 2018; You et al., 2018; Sharif et al., 2019). In addition, each of five patients had at least one variant in genes from these particular pathways. Among genes from overrepresentation gene sets were LRP6, FZD2, COL4A4, and WNT1.

Ocular cells, including corneal epithelial stem cells, express components of the Wnt/β-catenin signaling pathway during eye development (Nakatsu et al., 2011). The LRP6 gene encodes the low-density lipoprotein receptor-related protein, which is a component of a Wnt receptor complex. This complex is involved in Wnt ligands binding resulting in the nuclear translocation of β-catenin and regulation of the transcription of target

**Table 3 The expression of genes with rare variants identified in ES analysis.**

| Pathway | Gene symbol/ variant | KC_15 | KC_16 | KC_17 | KC_18 | KC_19 | Mean expression |
|---|---|---|---|---|---|---|---|
| Wnt Signaling | WNT1 c.1063G>T | 0.051 | 0.000 | 0.000 | 0.018 | 0.023 | 0.0103 |
| Focal Adhesion | PTK6 c.1066C>A | 27.705 | 31.413 | 22.991 | 25.809 | 26.031 | 26.5610 |
| Wnt Signaling | FZD2 c.1238G>A | 1.254 | 0.151 | 2.988 | 0.453 | 0.544 | 1.3098 |
| Wnt Signaling | VANGL1 c.41C>T | 4.655 | 3.581 | 4.129 | 3.132 | 5.074 | 4.2475 |
| Focal Adhesion | COL4A4 c.2347G>A | 8.802 | 8.487 | 15.100 | 5.863 | 7.398 | 9.2908 |
| Wnt Signaling | LRP6 c.4822C>T | 14.144 | 13.303 | 11.900 | 11.515 | 12.581 | 12.8858 |
| Wnt Signaling | PPP3CC c.1199T>C | 12.523 | 14.479 | 9.580 | 8.556 | 10.525 | 11.5208 |
| Focal Adhesion | BIRC2 c.1528A>G | 42.675 | 45.517 | 32.853 | 30.261 | 40.725 | 40.4425 |
| Focal Adhesion | PPP1R12A c.254A>G | 83.939 | 83.438 | 61.009 | 55.103 | 76.460 | 70.8723 |
| Focal Adhesion | PAK6 c.1855C>A | 8.818 | 7.144 | 5.704 | 4.970 | 6.011 | 6.6590 |

**Note:**
Grey color indicates the expression level in samples in which variant was present, and white color indicates the expression level of genes in samples without the presence of the analyzed variant.

genes (*Zhang et al., 2015*). Knock-out of β-catenin or its both co-receptors (*Lrp5* and *Lrp6*) in mouse corneal stromal cells resulted in premature stratification of the corneal epithelium, suggesting these genes play a role in the regulation of corneal morphogenesis (*Zhang et al., 2015*). In this study, the expression of *LRP6* in the cornea of the carrier of variant c.4822C>T (KC17) was unchanged compared to the other four KTCN individuals. However, since this variant was identified in KC17 patient in combination with another variant in the gene of Wnt signaling pathway (*PPP3CC*), we suggest that this variant might be a part of the specific combination of KTCN variants and its identification is not incidental.

The *FZD2* gene encodes frizzled class receptor 2, another protein functioning as Wnt ligands receptor and regulating the eye development through Wnt/β-catenin signaling. Xenopus frizzled-2, a homolog of the human frizzled receptor, is highly expressed in the embryo, including the developing eye (*Deardorff & Klein, 1999*). Interestingly, a significant increase of mRNA and protein expression of secreted frizzled-related protein 1 (SFRP1), which is a Wnt antagonist, has been detected in KTCN corneal epithelium and corneal buttons (*Sutton et al., 2010*; *You et al., 2013a*). In contrast, tear SFRP1 level has been significantly decreased in KTCN patients compared to control individuals (*You et al., 2013b*). The c.1238G>A variant in *FZD2* was identified in the cornea of KC16 patient in combination with c.41C>T variant in *VANGL1* (Wnt signaling) and c.2347G>A variant in *COL4A4* (focal adhesion). It is known that collagens are major

components of human corneas and the thinning of corneal stroma in KTCN may be the final effect of a disorganized collagen lamellae arrangement (*Meek et al., 2005*; *Mathew et al., 2015*). The *COL4A4* gene is expressed in human cornea (*Kabza et al., 2017*) and based on several studies it has been reported as a candidate gene for KTCN (*Stabuc-Silih et al., 2009*; *Wang et al., 2013*; *Kokolakis et al., 2014*; *Saravani et al., 2015*). However, genetic analyses of this gene in different populations have given ambiguous results about the role of *COL4A4* in KTCN development (*Stabuc-Silih et al., 2009*; *Wang et al., 2013*; *Kokolakis et al., 2014*; *Saravani et al., 2015*). While there was no difference in expression of *COL4A4* and *VANGL1* in the patient's cornea, the expression value of *FZD2* in the patient KC16 was lower compared to the expression observed in the other four KTCN individuals. However, further research should be performed to interpret the obtained data.

The combination of variants in *WNT1* and *PTK6* was revealed in the patient KC15. *PTK6* encodes tyrosine kinase 6 protein, which is involved in focal adhesion, as well as in GTPases and MAP kinases regulation. Signaling by PTK6 is implicated in controlling the differentiation of normal epithelium and tumor growth (*Brauer & Tyner, 2010*). However, there is no report about the role of *PTK6* in the maintenance of corneal epithelium.

Also, in KC18 and KC19 patients, variants in other elements (*BIRC2* and *PAK6* with *PPP1R12A*, respectively) of Wnt signaling and/or focal adhesion pathways were observed. These results further suggest that the alteration of genes regulating these signaling pathways might be an important risk factor for KTCN.

To determine the likelihood of seeing the overrepresentation of genes from Wnt signaling and focal adhesion in KTCN samples by chance, we performed analyses of exome data of five randomly selected Polish individuals without eye disease, as well as five Polish individuals with HM. Variants from control ES data were selected using the same filtering criteria as we used for the KTCN study, followed by enrichment analyses. No genes involved in Wnt signaling or focal adhesion pathways were overrepresented in our control sample sets, confirming that enrichment of variants in genes from these pathways in KTCN individuals was not incidental.

The small sample size disabled a reliable statistical analysis and due to this limitation, the general conclusions could not be made. However, the results of this pilot study gave additional insight into the role of the Wnt signaling and/or focal adhesion pathways in KTCN development and showed the possible indications for further KTCN research. The identification of rare variants in different genes further supports the heterogeneity of KTCN with multiple genes underlying its pathogenesis (*Karolak & Gajecka, 2017*; *Abu-Amero, Al-Muammar & Kondkar, 2014*; *Bykhovskaya, Margines & Rabinowitz, 2016*; *Mas Tur et al., 2017*; *Valgaeren, Koppen & Van Camp, 2018*).

## CONCLUSIONS

Summarizing, this first pilot ES profiling of human KTCN corneas indicates that the accumulation of variants in several genes from Wnt signaling and/or focal adhesion pathways might cause the phenotypic effect and further explain the involvement of these pathways in KTCN. Moreover, it also supports the hypothesis about the complex basis

of KTCN. Since five patients were evaluated, the role of variants in genes of these overrepresented pathways in KTCN etiology should be further elucidated in a larger group of patients using high throughput methods.

### Funding

This work was supported by the National Science Centre in Poland, Grants 2013/10/M/NZ2/00283 and 2018/31/B/NZ5/03280. The funders had no role in study design, data collection and analysis, decision to publish, or preparation of the manuscript.

### Grant Disclosures

The following grant information was disclosed by the authors:
National Science Centre in Poland: 2013/10/M/NZ2/00283 and 2018/31/B/NZ5/03280.

### Competing Interests

The authors declare that they have no competing interests.

### Author Contributions

- Justyna A. Karolak conceived and designed the experiments, performed the experiments, analyzed the data, prepared figures and/or tables, authored or reviewed drafts of the paper, and approved the final draft.
- Tomasz Gambin analyzed the data, prepared figures and/or tables, authored or reviewed drafts of the paper, and approved the final draft.
- Malgorzata Rydzanicz performed the experiments, analyzed the data, authored or reviewed drafts of the paper, and approved the final draft.
- Piotr Polakowski analyzed the data, authored or reviewed drafts of the paper, and approved the final draft.
- Rafal Ploski analyzed the data, authored or reviewed drafts of the paper, and approved the final draft.
- Jacek P. Szaflik analyzed the data, authored or reviewed drafts of the paper, and approved the final draft.
- Marzena Gajecka conceived and designed the experiments, analyzed the data, authored or reviewed drafts of the paper, and approved the final draft.

### Human Ethics

The following information was supplied relating to ethical approvals (i.e., approving body and any reference numbers):

The study protocol was approved by the Institutional Review Board at Poznan University of Medical Sciences (453/14 and 755/19).

### DNA Deposition

The following information was supplied regarding the deposition of DNA sequences:

The data are available in the ClinVar database: SUB3758236 and SUB4926771.

## Data Availability

The raw data are provided in Table S2.

## Supplemental Information

Supplemental information for this article can be found online at http://dx.doi.org/10.7717/peerj.8982#supplemental-information.

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
