# Peer review of "Accumulation of sequence variants in genes of Wnt signaling and focal adhesion pathways in human corneas further explains their involvement in keratoconus"

_PeerJ, doi:10.7717/peerj.8982_

## Round 0.1 · original submission · Major Revisions

Please carefully address all critiques of both reviewers and revise your manuscript accordingly.

Reviewer 1 ·

Basic reporting

This study provides an exome sequencing approach to study a small series of keratoconus-affected corneas. This is a pilot study and should be noted as such. Can the authors comment on the incidence of keratoconus in the Polish population and how this compares with other ethnic groups (for example, Maori, Indian).

The manuscript is quite well written but does require editing for grammar and expression in places.

The Abstract does not have a conclusion but assume this is encompassed by lines 36 to 39.

The Introduction provides some overview of the study background but does miss some key references related to RNASeq in keratoconus (for exmaple, Sharif et al., 2019; Khaled et al., 2028; You et al., 2018). The expression of Frizzled proteins in keratoconus has also been reported in earlier studies (for example, You et al., 2013; Sutton et al. 2010).

There have been several GWAS studies in keratoconus (briefly referred to in lines 64 to 67); a few candidate genes are mentioned but there is limited detail - please provide further references and context.

The study aim (lines 68 to 70) indicates it was “to assess involvement of new genetic variants in KTCN etiology ……of Polish patients with KTCN, previously tested by RNA-Seq, using exome sequencing (ES) approach.” The authors also suggest this is the first exome sequencing study for keratoconus (Abstract lines 36 to 37, Conclusion line 218), however whole exome sequencing for keratoconus has been published previously (for example Magalhães et al., 2019; Karolak et al., 2016, Lucas et al., 2018).

Experimental design

The methods for DNA preparation and gene analysis seem appropriate with patient consent and institutional ethics approval. A major issue that requires further comment is the limited sample size (n=5 keratoconus corneas), and whether the data are thus sufficient to support the manuscript title and the conclusions.

Furthermore, the authors note that DNA from an additional 5 control and 5 high myope corneas was also analysed, however no details are provided for these samples (line 122 to 125). Can the authors clarify that the corneas analysed for DNA were all late stage (just before penetrating keratoplasty surgery)?

The DNA extraction from the keratoconus (and other?) corneas was presumably from whole corneal tissue (i.e. all layers); please state this in the Methods. It may also be worthwhile for the authors to assess gene variants using exome sequencing of the individual corneal layers.

The authors note line 157, that “.. data were deposited in the freely accessible ClinVar database (submission ID:158 SUB3758236)”. Please indicate if the control and myopia gene data is also available (this was not obvious from the ClinVar Submission).

Validity of the findings

As noted in the Discussion limitations, this is a pilot study (line 214) and the authors should clearly state this in the title and abstract, as well as in the Discussion (line 214).

The findings show that there are rare gene variants involved in late stage keratoconus and support the complex and heterogenous nature of keratoconus; bothese concepts have been reproted previously. The involvement of the Wnt/frizzled pathways has also been reported previously (see earlier comments).

Validation of the genes highlighted from the exome sequencing approach, using either protein or qPCR approaches, would enhance the current findings.

·

Basic reporting

No comments

Experimental design

No comments

Validity of the findings

No comments

Additional comments

The manuscript is well written in professional language, and experiments are well-designed. There are some suggestions to improve the manuscript.

1. In Table 1, pathways listed are redundant, e.g., several Wnt pathways are listed. Please simplify.
2. Table 2, please include variants information, whether it is synonymous or non-synonymous.
3. Table 3 is comparing gene variants expression between patients’ samples. Please also include information from controls.
4. In Supplementary Table 1: it would be great to include the information of an average corneal thickness (Pachymetry).
5. Some sentence are too long. It is easier for understanding if split into two sentences. For example, in the abstract, “in each of five patients, the assessment of functional interactions revealed rare …….
6. Please cite supplementary Table 2 in your manuscript, probably in line 134 after ‘ES screening of KTCN corneas revealed 117 potentially relevant variants with MAF ≤ 0.01 in….’.

---

## Round 0.2 · accepted · Accept

Since all critiques were addressed and the manuscript was adequately amended I am glad to accept revised version.

·

Basic reporting

This revised manuscript is much improved and is suitable for publication .

Experimental design

No comments

Validity of the findings

No comments

Additional comments

1. Line 174, unify the use of numbers, ‘nine’ should be ‘9’
2. Line 189, change ‘has’ to ‘have’
3. Line 190, should be ‘genetic variants’